# From Acts of Care to Practice-Based Resistance: Refugee-Sector Service Provision and Its Impact(s) on Integration

Emmaleena Käkelä *, Helen Baillot, Leyla Kerlaff and Marcia Vera-Espinoza 

Institute for Global Health and Development, Queen Margaret University, Edinburgh EH21 6UU, UK
* Correspondence: ekakela@qmu.ac.uk

**Abstract:** The UK refugee sector encompasses welfare provision, systems advocacy, capacity development and research. However, to date there has been little attention on refugees' experiences of the support provided by these services or on the views of the practitioners who deliver them. This paper draws from interviews and workshops with thirty refugee beneficiaries of an integration service in Scotland and twenty practitioners to shed light on how refugees and practitioners perceive and provide meaning to the work of the refugee sector. We identify refugee sector organisations as crucial nodes in refugees' social networks and explore the multiple roles they play in the integration process. Firstly, we confirm that refugee organisations act as connectors, linking refugees with wider networks of support. Secondly, we demonstrate that the work of the refugee sector involves acts of care that are of intrinsic value to refugees, over and above the achievement of tangible integration outcomes. Finally, we demonstrate that this care also involves acts that seek to overcome and subvert statutory system barriers. We propose to understand these acts as forms of "*practice-based resistance*" necessitated by a hostile policy environment. The findings expand on understandings of the refugee sector, its role in integration and the multi-faceted nature of integration processes.

**Keywords:** refugee-sector; refugee integration; practice-based resistance; care; hostile environment; welfare restrictionism

## 1. Introduction

Global events and national policy changes have had a major influence on the UK third sector. Charities have arguably played an important role in addressing social needs even prior to the emergence of the contemporary British welfare state (Harris and Bridgen 2012). For asylum seekers and refugees, third sector organisations are often the first providers of support (Calò et al. 2021). In particular, refugee community organisations have been at the frontline in addressing the progressively worsened welfare disentitlement and social exclusion of asylum seekers and refugees in the UK which has been taking place since the mid-1990s (Zetter and Pearl 2000). It has been argued that the increasingly restrictive UK asylum policy regime has created a rising demand for third sector welfare services; since the 1990s there has also been a substantial increase in the number of charities supporting destitute asylum seekers and refugees (Mayblin and James 2019). Recently, similar trends have also been witnessed elsewhere in Europe, where non-state actors have played a crucial role in addressing the challenges of the so-called "refugee crisis" and filling the gaps in statutory service delivery (Galera et al. 2018; Garkisch et al. 2017). During the last decade of austerity, the UK refugee sector has been pressed to meet this rising demand amidst constrained funding (Mayblin and James 2019).

This article draws from our practice-research partnership with established refugee-sector organisations which partake in both integration service delivery and policy advocacy in Scotland[1]. After presenting an overview of the UK policy and practice context and gaps in the literature, we briefly discuss ideas around integration and care, to then outline our research methodology and mixed-methods approach. This has been based on practice-research engagement (PRE) bringing together researchers, practitioners and service users

to facilitate mutual learning and positive social change (Brown et al. 2003). Refugees' own descriptions of their experiences with the UK refugee sector are foregrounded. We then contrast refugees' experiences with practitioner insights to further reflect on the role and functions of the refugee sector, developing the notion of "*practice-based resistance*" as an integral dimension of integration service delivery. We contribute new empirical knowledge to both third sector and integration studies about the role of migrant organisations in integration processes from the seldomly heard perspectives of integration service practitioners as well as refugees. These insights add to understandings of the provision of integration support provided by the third sector, highlighting not only the type of support that is provided, but also the way in which this support is experienced. This offers an opportunity to expand our conceptual understanding of the multi-faceted nature of integration processes and the role of care and resistance within these.

## 2. Immigration & Asylum Policy Context

The UK asylum system has been characterised by a culture of disbelief (Käkelä 2022), scaffolded by policies which have progressively furthered the racialisation and criminalisation of asylum seekers (Bhatia 2020; Farmer 2021). In 2012, the then Home Secretary Theresa May announced that the UK Government would openly pursue a hostile environment policy to tackle irregular immigration (Griffiths and Yeo 2021). Her intervention was linked to attaining the then government's net migration target, bringing down numbers of arrivals that were perceived to have reached unsustainable levels, posing a threat to social cohesion (Casey 2016). However, the foundations of the hostile environment trace further back; under New Labour Governments since mid-1990s, consecutive legislation was introduced to restrict the rights of asylum seekers, including the exclusion of asylum seekers from mainstream benefits, housing and the labour market, detention without time limit and increased powers to remove migrants (Stevens 2001; Saunders and Al-Om 2022). The hostile environment policy is driven by strategies of deterrence which restrict irregular migrants' access to essential services and "deputise" immigration control to statutory service providers through compulsory data sharing with the Home Office (Griffiths and Yeo 2021; Bhatia 2020). Although the UK Government has since then opted to describe its policy in terms of compliance environment (Saunders and Al-Om 2022), the substance of the policy approach remains largely the same. Most recently, the Nationality and Borders Act 2022 has created a two-tier system which grants asylum seekers differing rights and access to essential services based on their route of arrival, against mounting criticism across the legal, human rights and refugee sectors (Refugee Council 2022). These developments have taken place despite wealth of evidence showing that the exclusion of asylum seekers from public funds causes destitution, exacerbates intersectional inequalities and perpetuates physical, psychological and gendered harms (Bhatia 2020; Flynn et al. 2018; Canning 2017), at the same time doing little to reduce the number of new asylum applications (Griffiths and Yeo 2021).

In the UK, integration policies represent an area of policy divergence; while the central government has adopted a stance that integration only begins when refugee status has been granted (Mulvey 2015), the devolved Scottish administration has championed integration "from day one" through its own integration strategy (Scottish Government 2018, p. 6). European funds have been allocated by the Home Office to support migrant organisations to deliver integration services to status refugees. Nonetheless, "failures to integrate" continue to be laid primarily at the door of newcomers rather than wider societal structures (Phillimore 2021). Scotland receives asylum seekers through the UK policy of dispersal. This centralised system allocates people who are seeking asylum and who, because they would otherwise be destitute, require accommodation and financial support, to dispersal regions around the UK on a "no-choice" basis according to availability of bed spaces. Until the Ministerial announcement in April 2022 that all local authorities would be expected to participate in asylum dispersal[2], Glasgow had been the only dispersal city in Scotland. Scotland has also received refugees and other displaced people through UK

Resettlement Schemes and most recently the Homes for Ukraine Scheme, but these routes are beyond the scope of our paper.

The Scottish *New Scots Refugee Integration Strategy* has been informed by the *Indicators of Integration* framework, which conceptualises integration as a multi-directional, multi-dimensional and context-specific process (Ndofor-Tah et al. 2019). The Framework was commissioned by the UK Home Office; the original version was developed by academics (Ager and Strang 2004) and later updated in collaboration with the Home Office and a wider team of academics (Ndofor-Tah et al. 2019). The framework brings together *Markers and Means* (housing, employment, education, leisure and health), *Social Connections, Facilitators* (culture, language, safety, stability and digital skills) and rights and responsibilities which provide a *Foundation* for integration. Nevertheless, there are notable differences in how the two governments champion these outcomes. In contrast to Westminster, the Scottish Government grants asylum seekers access to further education, free secondary healthcare regardless of status, and flexibility for recognised refugees to apply for social housing outwith the local authority area where asylum seekers are initially housed (Scottish Government 2018; British Medical Association 2022). However, critics have argued that immigration policies reserved to the UK Government, including no-choice asylum dispersal and removal of the right to work for most asylum seekers, have negated the Scottish Government's capacity to promote positive integration outcomes (Mulvey 2015). With the recently passed Nationality and Borders Act 2022 which legitimises a two-tier approach of provisions for asylum seekers depending on their route of arrival, it is likely that the devolved administrations will face increased challenges in turning "integration from day one" from an aspiration into a reality for asylum seekers in Scotland.

## 3. UK Refugee Sector

There are considerable country variations in both policy and legal contexts of refugee integration and the role of third sector in facilitating the integration of new refugees (Galera et al. 2018). In contrast to countries like Finland where refugee integration is formally recognised in law as a dimension of welfare provision led by local authorities (Finlex 2010), refugee integration in the UK has been spearheaded by a strong third sector. While other statutory and community organisations may work to support refugees, this article focuses upon the role of third sector organisations whose primary remit is to work with refugees and people seeking asylum. For the purposes of our analysis, we use the term refugee sector to describe these organisations, whilst cognisant that the sector is not a homogenous entity. We are conscious too that much of our analysis focuses on the work of well-established and relatively large charitable organisations rather than on grassroots work such as that undertaken by refugee community organisations (Zetter and Pearl 2000). However, we have, wherever possible, incorporated data reflecting the importance of less formalised integration support and of refugees' own capacities to support themselves and their peers.

International research has identified four domains of provision through which refugee-sector organisations respond to challenges of migration, namely: provision of basic services and social welfare; capacity development; system-oriented advocacy; and related complementary research activities (Garkisch et al. 2017). EU research has identified: embeddedness in local communities; a dual approach to empower both newcomers and local communities; and strong levels of trust with public, private for-profit and non-profit actors as key features of third sector organisations which support refugees' pathways to integration (Galera et al. 2018). This resonates with the UK refugee-sector which offers varied forms of support; from immediate advocacy and access to emergency essentials, to long-term advice, policy influencing and provision of opportunities, including volunteering. Previous research focusing on the UK context has addressed obstacles faced by community organisations and refugee organisations arising from the UK political and economic climate (Calò et al. 2021; De Jong 2019; Griffiths et al. 2006) and sector workforce (De Jong 2019). Research has also examined: refugee-sector employability services; (Calò et al. 2021) contexts which can be conducive for the emergence of refugee supporting organisations (MacKenzie et al. 2012);

the distinctiveness of small informal community organisations (Phillimore et al. 2010); and the scale of destitution support offered by registered refugee sector charities (Mayblin and James 2019).

Reliance on third sector organisations is pronounced among asylum seekers, almost all of whom are excluded from the labour market with no recourse to public funds. The long uncertain wait for asylum caused by backlogs in asylum decision-making, rate of initial refusals and resulting risk of destitution (Migration Observatory 2021) can have longstanding effects on refugees' capacity to move forward after being granted Leave to Remain. Even with relatively quick cases, newly recognised refugees often experience significant barriers in accessing rights, services and the UK labour market, which service providers must step in to navigate (Strang et al. 2017). These challenges are further compounded by funding constraints, which have limited refugee-sector organisations' capacity to tailor their services to users (Calò et al. 2021). Somewhat paradoxically, although the third sector has been expected to rise to meet these challenges, much of the asylum estate, security and support provision have been privatised (Calò et al. 2021). The move to contract a single third sector operator (Migrant Help) for the provision of asylum support services by the Home Office in 2014 led to substantial loss of funding among other refugee-sector providers (De Jong 2019). By the terms of its contract, Migrant Help must not take any actions which would adversely affect the Home Office, including providing advocacy for their clients (Asylum Matters 2019; De Jong 2019). In contrast, one of the previously identified strengths of the refugee sector has been determination to act independently to address the needs of communities which are often neglected by policymakers (Galera et al. 2018). The wider sector has been characterised by compassionate action, and a "culture of doing more with less as they help refugees to rebuild new lives" (Hack-Polay and Igwe 2018, p. 9). However, this compassionate approach is not wholly unproblematic, as it has been argued that "asymmetrical relation of 'giving' [ . . . ] may, in fact, replace care with charity" (Darling 2011, p. 411).

Outside of the academic realm, service evaluations and reports confirm the centrality of the support offered by the sector at different transition points in refugees' lives (Baillot et al. 2016; British Red Cross 2022). There is evidence too of the powerful nature of collaborative work between refugee sector organisations who work to enact strategic and policy change (Stop Lock Changes Coalition 2019). More recently, this has been demonstrated by the collective organising through the Together with Refugees[3] campaign to oppose the recent Nationality and Borders Act 2022, solidarity protests against dawn raids (Goodwin 2021), and sector-wide advocacy which led to the European Court of Human Rights (ECHR) intervention to halt a flight destined to remove asylum seekers to Rwanda (Leigh Day 2022). While evidence of the functional value of the refugee sector therefore abounds, less attention has been paid to the manner in which the daily work of practitioners is experienced by refuges. This paper addresses this gap by putting in dialogue the insights of both asylum-route refugees and practitioners, while mobilising ideas around care and resistance within processes of integration.

## 4. Social Connections and Integration

Despite, or perhaps because of its widespread adoption beyond the realms of academia, the use of the term integration as a conceptual underpinning for research with refugees has been problematised (Spencer and Charsley 2021; Schinkel 2013). Some scholars have called for it to be rejected, positioning it as being inherently normative and relying upon racialised perceptions of host societies and cultures (Schinkel 2013). These, and similar critiques have been summarised by Spencer and Charsley (2021), who suggest that researchers should retain the concept whilst addressing the critiques they identify. This is the approach we have adopted, and we situate our analysis firmly within an understanding of integration as a process that is influenced by time and context, rather than a normative and pre-determined set of outcomes (Penninx 2019).

Scholarship has increasingly highlighted the centrality of social connections in processes of integration (Strang and Quinn 2021; Phillimore 2021; Pittaway et al. 2016). Different types of social relationships, including those with people like you (most often conceptualised as bonds), those with people who are different to you (bridges) and those with organisms of the state (links) can simultaneously be indicators of integration and facilitators of it (Ndofor-Tah et al. 2019). Relationships with others reduce isolation (Strang and Quinn 2021), enable access to rights, information, services and employment (Gericke et al. 2018) and can contribute to changing social attitudes amongst refugees and receiving communities (Daley 2009). While using static categorical distinctions to distinguish between different types of relationship in post-migration contexts has increasingly been contested (Wessendorf and Phillimore 2019), insights from this scholarship are found in the UK and wider European integration strategies, which privilege socio-economic integration (Ndofor-Tah et al. 2019; Scottish Government 2018; Scholten et al. 2018). Nevertheless, there is a lack of evidence-based understanding on how to promote refugees' social connectedness through practice and policy measures. Our research has sought to address this gap, and in doing so has revealed that migrant organisations play multiple roles not only as social connectors but also as social connections in their own right. It is the affective role played by these relationships between asylum route refugees and migrant organisations that we focus on in this paper, characterised and defined by an ethics of care and shared humanity (Caduff 2019; Scuzzarello 2009). Care comes to the fore in part at least due to the constraints imposed by the wider policy environment surrounding work with refugees.

## 5. The Politicisation of Care

Care has been mostly understood in relation to a wide range of tasks, activities and practices "to promote the personal health and welfare of people who cannot, or who are not inclined to, perform those activities themselves" (Yeates 2004, p. 371). Within migration and refugee studies, debates around care have included approaches that explore feminist care ethics with some authors interrogating caring activities migrant women take professionally and within private spaces (Kofman and Raghuram 2015), as well as the racialisation of these practices (Raghuram 2019). There are also debates that interrogate the state's approach towards refugees by understanding care as an "essential ethical attitude in the recognition of every human being", beyond conditional ideas around solidarity (López-Farjeat and Coronado-Angulo 2020, p. 9).

While there is a wide range of literature on the moral and political philosophy of care which goes beyond the scope of this paper, it is key to note that most discussions about care have shed light on the relevance of both context and connection (Scuzzarello 2009) as well as understandings of care as a value and a practice (Raghuram 2016). White and Tronto (2004, cited in Raghuram 2019, p. 8) argue that there are four interconnected pillars to care that link these values and practices: *attentiveness* related to *caring about* and the need for caring; *responsibility* related to *caring for*, that is who should assume the responsibility to meet the needs for care; *competence* related to *care giving* and the performance of a necessary caring task; and *responsiveness*, which relates to *care receiving* or the extent to which care meet the needs of the cared for and the carer. These discussions of care shed light on the place where care is given and where it is received (Raghuram 2016). However, some authors recognise that an individual position as caregiver or care-taker can also shift in time and space (Scuzzarello 2009).

Drawing on this relational understanding of care, in this paper we put focus on how care is *perceived* and *experienced* by both caregiver and care-receiver within processes of integration in a context of restrictive immigration policies. We thus pay attention to care as "a scene of intimate connection" (Caduff 2019, p. 788), with multiple purposes that include the expression of solidarity as a feature of institutional and professional life for those who provide it (Caduff 2019), and the affective and functional role perceived by those who receive care. In his discussion of how care can be understood, Caduff hints at the importance of situating care giving within its political and social, as well as familial context, while also



problematising that "care is often difficult for those who require it, those who receive it, and those who provide it" (Caduff 2019, p. 789). Others have gone further and called for care to be more explicitly politicised. As Emejulu (2018) explains: "care about others is not mere empty empathy. To care about others requires the development of a political imagination that takes seriously the lived experiences of the most marginalised". This politicisation of care goes beyond the moral value of care and engages in the possibilities of the practice of care to *challenge* the hostile environment as a form of *resistance*. Drawing on Saunders and Al-Om (2022); see also (Lilja and Vinthagen 2018), the paper understands resistance as acts that challenge power relations of some kind. In this case, we discuss daily acts and practices of care that are deployed by the refugee sector, which can challenge and undermine the power of the hostile environment, while at the same time facilitating refugees' inclusion. Thus, in this paper, we engage with both perceptions of care (in relation to the value that is attached to it), and also the politicisation of care as resistance (related to the strategic practice of it) in order to understand the role that the refugee sector plays in pathways to inclusion as a key social connector, and the role that care—as experienced by refugees—can play in integration.

## 6. Methodology

This paper draws from an ongoing study on adult refugees' pathways into social and economic inclusion in Scotland funded by AMIF (2020–2022). This was a participatory collaboration with three well-established refugee-sector organisations which deliver an Integration Service Partnership comprised of individual advice and integration planning (organisation 1), employability support (organisation 2) and English language provision (organisation 3) to recently recognised refugees. Two of the partnership organisations cater their services solely to refugees and asylum seekers, while the third partner also delivers adult learning opportunities to other cohorts. Two out of the three organisations employ staff and volunteers from refugee backgrounds, and in the case of the lead partner, staff with refugee backgrounds are represented at the highest levels of the organisation. Additionally, organisation 1 also delivers a Peer Project, which is a refugee-led initiative to facilitate sharing of experiences and learning around rights, responsibilities and services.

Our partnership can be characterised as practice-research engagement, which seeks to combine practice insights with analytical tools of research to produce new knowledge and benefits to practice (Brown et al. 2003, p. 83). The role of the research team in this collaboration has been twofold; firstly, to build knowledge on the role of social connections in refugee integration; and secondly, to enhance service delivery through embedding research-based tools into practice to map casework impact into refugees' social connections. Our collaboration has centred around dialogical process of knowledge production to both feed research insights into practice developments and to continuously engage with practitioners and refugees to embed their feedback in the development and refining of the research focus and methods, and the analysis of the data. Before data collection conducted in 2021–2022, the research was granted ethical approval by the Queen Margaret University Ethics Committee (REP 0244).

The findings reported in this paper are based on a sub-sample of six individual interviews and seven focus group discussions conducted during the course of the study, with a total of 30 asylum-route refugee participants. This includes 15 men and 15 women of different nationalities. A further 10 individual interviews were excluded from this round of analysis due to timings and the specific focus of these interviews on social connections that help refugees to access and maintain secure housing. Three of the focus group discussions were with refugees who were also volunteering for the Peer Project run by organisation 1. Participants were purposively sampled for the remaining activities from across the elements of the partnership's services to reflect a range of experiences of beneficiaries currently engaged with the integration service. All participants were recruited through the three partner organisations and broadly reflect a mix of genders, nationalities and English language proficiencies. Whilst we did not purposively sample on the basis of

religion or age, participants were from a range of faith backgrounds and the majority were broadly estimated to be between the ages of 25–45, with very few older participants. All participants were provided with information about the research by their service provider and subsequently by members of the research team through translated materials including written and verbal information and short videos. The sample of participants includes refugees with different language levels; the focus groups were conducted in English, whereas individual interviews were done with the aid of interpreters. Most of the data collection was conducted online through Zoom and Microsoft Teams but following the easing of the Scottish COVID-19 restrictions, we were also able to engage with seven participants face-to-face in partner organisation premises in Glasgow. Due to the scope of funding to deliver the Integration Service Partnership, all participants had been recognised as refugees excepting three of the peer volunteers who were still waiting for a decision on their asylum cases. Additionally, this paper draws from data gathered from 20 refugee-sector practitioners who participated in four online focus groups during our practice-research engagement.

The research utilised in-depth semi-structured individual interviews, focus groups and a quantitative social connections mapping survey, last of which is beyond the scope of this paper. The interviews and focus groups embedded a visual mapping method which was inspired by previous research which sought to define social capital from a refugee perspective through community consultations (Pittaway et al. 2016). In our research, we utilised a relational network mapping exercise developed by Pittaway et al. (2016) to identify what people and organisations had helped refugees in settling in Scotland, and to facilitate discussions on the functions and meanings of these connections. During the data collection, participants were invited to reflect on the question of "what people and organisations have been important to you in your life in Scotland?". The named connections were visualised into a bullseye board using Miro Mind Mapping Tool (see Figure 1), or in the case of face-to-face workshops, on A3 sheets using Sticky Notes. The researchers followed with further probes on how and why named people and organisations had been helpful to the participants, how they had made these connections and whether there had been connections they still wanted to develop. Participants were also asked to identify which of the connections had been the most important to them—these were then placed on the inner circle of the bullseye. In our research, the bullseye offered a visual tool to facilitate participant memory and in the case of focus groups, sharing of experiences between the participants.

In the next sections, we will first report the findings of a quantitative content analysis of the bullseye maps, followed by a discussion on qualitative findings from inductive thematic analysis of the interview and focus group transcripts. The two analysis approaches were combined for the purposes of *triangulation* to seek corroboration between qualitative and quantitative approaches (Bryman 2006). The quantitative content analysis allows us to identify patterns of which people and organisations had been important to participants and which had been *most* important. The thematic qualitative analysis contextualises this data and elucidates *why* refugees' identified refugee service providers as crucial connections, and the meanings both refugees and practitioners attach to their relationship. After verbatim transcription, the pseudo-anonymised qualitative transcripts were first coded with the aid of Dedoose software, after which these codes were refined into themes. The next sections outline our key findings, followed by the discussion problematising their implications on how we conceptualise refugee-sector support. Participants' words are reported using pseudonyms selected by the research team to ensure confidentiality.

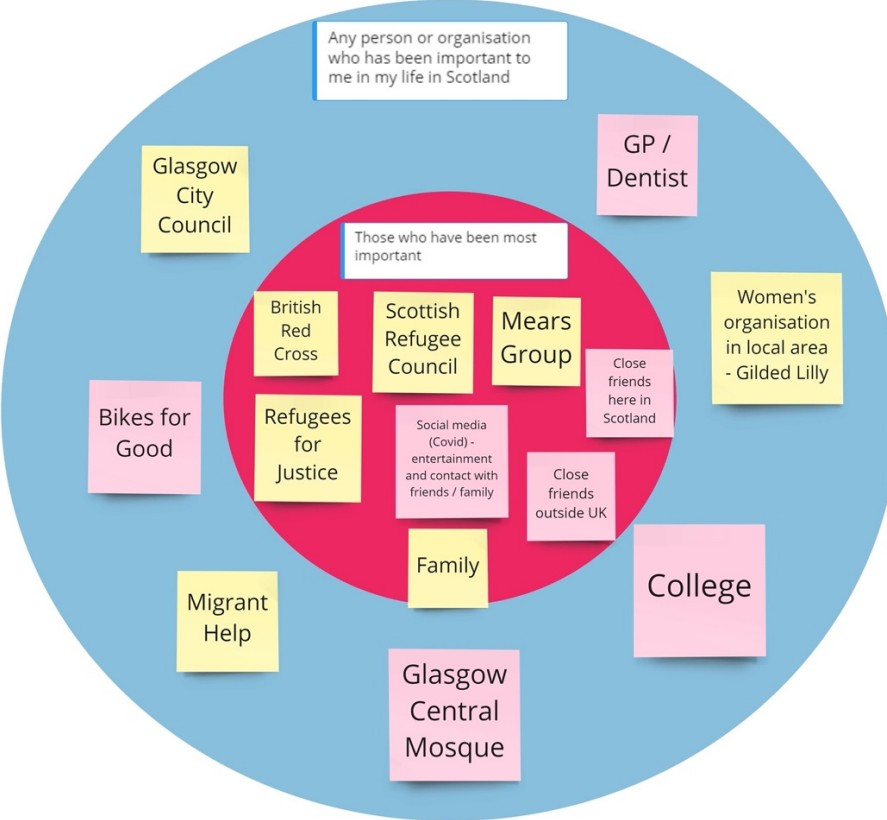

**Figure 1.** Bullseye map created with Miro Whiteboard Tool.

## 7. Findings

The qualitative analysis presented in the following sections was guided by an initial quantitative content analysis of the bullseye maps (Figure 1) created during the individual interviews and focus groups. Each connection was only labelled on the bullseye maps once per focus group, regardless of how many focus group participants mentioned the named connection as important to them.

The analysis counted the named important connections based on how many times these were mentioned in all sessions, to identify patterns in organisations and individuals which had played a particularly prominent role in participants' integration journeys. The most populated bullseye map identified 26 connections and the least populated map included 9 connections. These counted connections from 13 bullseye maps were then distilled into predetermined categories based on our pre-existing knowledge about the statutory and third sector welfare provision in the UK (see Figures 2 and 3). We counted both all mentioned important connections (Figure 2) and all connections named as being *the most important* to participants (Figure 3). As the bullseye maps represent connections mentioned per focus group, rather than per participant, the figures presented here do not allow comparisons made between demographic groups, which we instead sought to explore through the qualitative analysis of transcripts that give further context to individuals trajectories of integration. As reflected in later sections, several participants said that all connections had been equally valuable to them, which explains the discrepancy in counts presented in Figures 2 and 3.

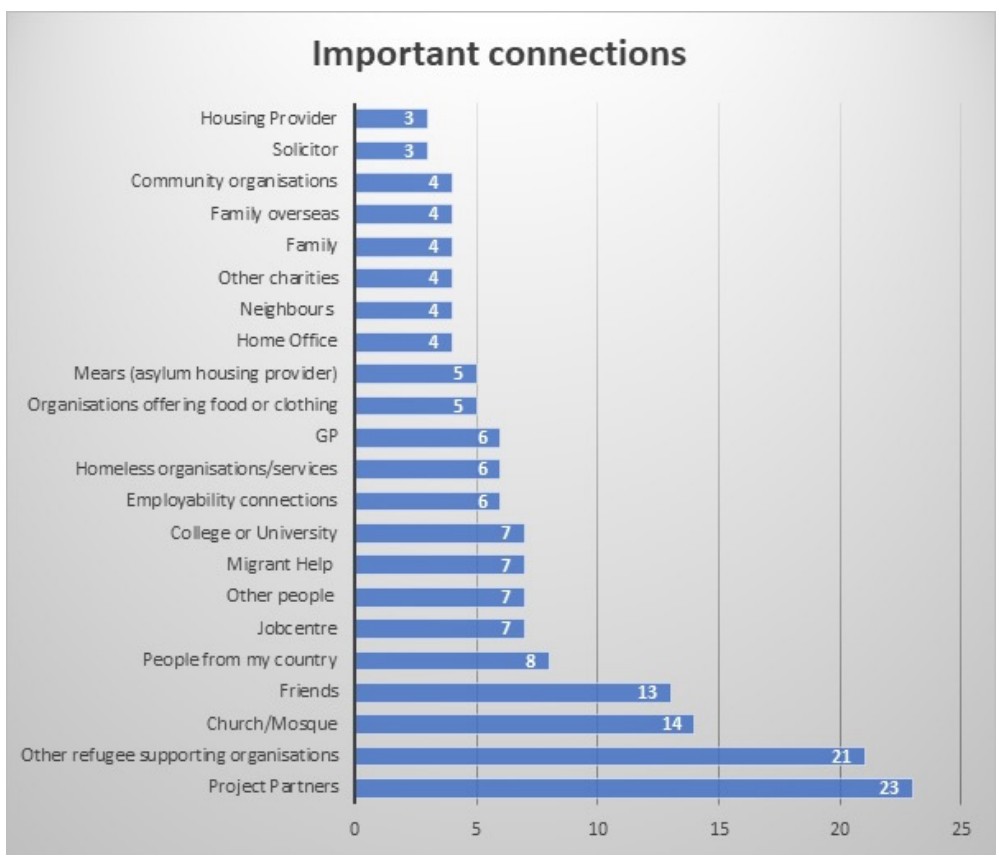

**Figure 2.** Important connections *based on how many times these came up* in bullseye conversations, regardless of how important these were stated to be.

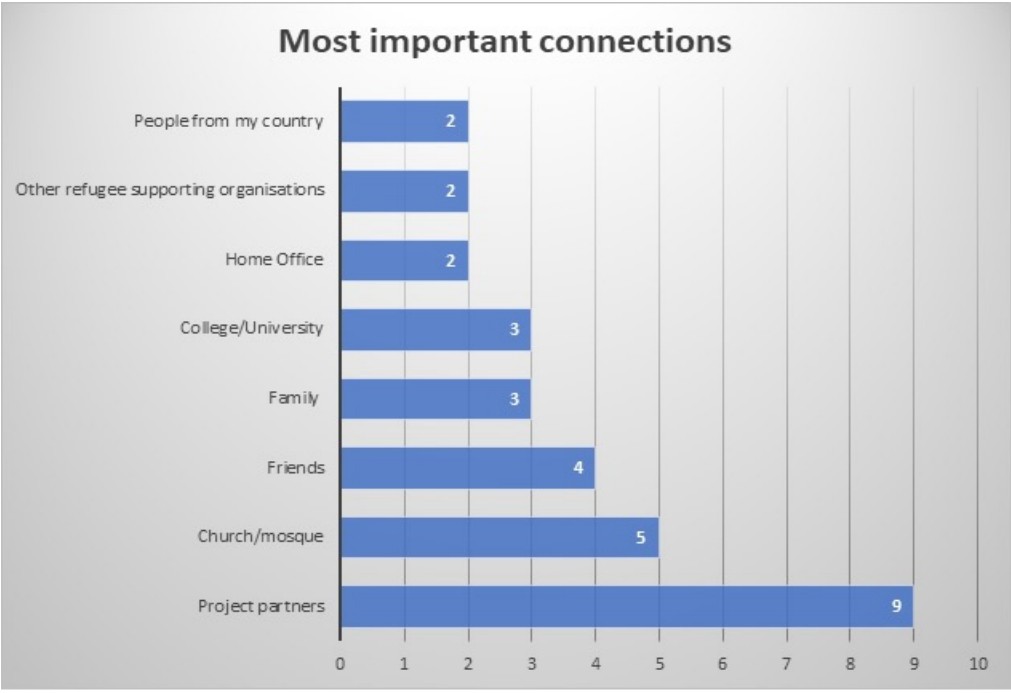

**Figure 3.** Most important connections *based on which connections were specifically stated as being the most important.* The figure only includes connections which were mentioned at least twice.

Notably, this initial analysis of patterns revealed that the Integration Service partners (labelled as project partners, Figure 2) and other community-based refugee-led organisa-

tions (labelled as other refugee supporting organisations, Figure 2) ranked highest in terms of their importance to refugees' lives in Scotland. When asked about *the most important* connections, Integration Service partners still ranked the highest. These organisations were not only mentioned most often but also often as the first connections when prompted by the initial question. These findings informed our qualitative analysis to examine the meanings and experiences of support from Integration Service partnership organisations and the wider refugee sector. The relatively small sample size, and the targeted recruitment through the partner organisations (which feature as the top connections for participants who were their service beneficiaries) means that these findings cannot be generalised to the wider refugee population in Scotland. Nonetheless, we consider that the findings presented in this paper highlight patterns which reflect the central role of the refugee-sector as an essential social connector and a provider of welfare and advocacy. The next sections illustrate qualitative findings on refugees' experiences of refugee-sector integration service provision as well as the views of the practitioners who deliver them.

### 7.1. Connecting: Building a Network of Support

Participants' perspectives confirm that rather than being singular entities, refugee sector organisations function as part of wider network of services supporting a range of needs. Several participants, when asked to rank the various people and organisations who had been important to them since arriving in Scotland, declined to do so as their importance was different but equal:

> *All of them have been equally important. All of them, inside me all of them, I cannot value more than another one because they have support me in different ways.* (Diego, refugee participant)

> *They're all important … Each has played their role in my life in the UK.* (Janine, refugee participant)

Participants shared their experiences of how refugee-sector organisations had been central in encouraging and enabling them to access mainstream services like colleges and libraries. The emphasis on the refugee sector as a node in a wider network was evident in many participants' experiences of being referred or signposted to services to address specific areas of integration, including language classes and support to navigate the process of family reunion. The Integration Service Partnership had meant that many of the participants had been proactively signposted from advice services to English language learning and specialist employability support, which for some had led to permanent employment or opportunities for further study.

In addition to connecting refugees with services and systems, refugee-sector organisations had also been a place to connect with others. Participants described having made friends in English classes [organisation 3] and having opportunities to engage with others through taking part in the Peer Project [organisation 1]. Crucially, for many refugees, it is not just links with formal organisations, but the informal social connections facilitated by them which are conducive to integration:

> *That's why I raise organisations who are willing to help refugees. When you're there, you will share information or you will tell people your background, what you want to do, what are your plans, and people are always there to help you to somehow help you to meet your needs or to help you achieve your goals. They get information from me, I get information from them, and with that information I just do everything.* (Aaron, refugee participant)

The functional role of informal connections was also emphasised by Integration Advisers, who recognised the ways friendships and acquaintances could open up unexpected pathways to further socio-economic inclusion. The value of informal, peer-to-peer social connections has also been embedded in some elements of the partnership, most notably through the Peer Project delivered by organisation 1 which sought to harness the knowl-

edge, experiences and skills of well-established refugees to provide support to more recent arrivals.

*7.2. Caring: Offering Kindness and Warmth*

For many participants, not only organisations but individual staff members had been an important part of their lives in Scotland. Individual caseworkers had not only helped with practical matters and advocacy, but were named as important sources of emotional support who had contributed to refugees regaining their confidence:

> *And the best thing I can describe [name of the caseworker, Organisation 1] is she's encouraging and also kind of worked on my self-esteem because when you're in a different country, you kind of are more hesitant, I guess. So that was like a brilliant experience. And I think all I'm doing right now is just, in effect . . . like a, resulted from her hard work.* (Miriam, refugee participant)

> *What's nice about what the classes [Organisation 3] teach you, how to speak, how to read, and these are all equally important . . . . . . Language is the basis for a social life, so when you have the language, you're much more confident anyway.* (Mahmoud, refugee participant)

For many participants, this confidence, once regained, had been an enabler for them to pursue their goals in other areas of life. Integration Advisers agreed that part of their work was about moving beyond being a mechanical advice service to building a genuine "human connection". This was also experienced by refugees themselves, and resulted for some participants in their feeling that their relationships with practitioners were familial rather than professional in nature:

> *The children, too, my son, he likes going to the office to see them. He sees them as an auntie and uncle [laughs].* (Aisha, refugee participant)

For refugees who felt otherwise alone in Scotland, this relationship mitigated their social isolation and so played a role unrelated to other domains of integration. Rather, the connection had value in and of itself, as a way to alleviate their sense of loneliness and solitude:

> *I didn't have anybody in Glasgow, it was just me. So, they come for visits, they call me, they encourage me to come to the office, so it was like a second home as well. I like seeing them. And when they have programme, family programme, we are invited to join in.* (Aisha, refugee participant)

The same participant described her employability support worker at organisation 2 as a "good friend" whom she could trust and who would always give her good advice. Likewise, even as her other employability worker in organisation 1 had moved onto a different field, she felt she could still reach out to her for help. This is illustrative of the strength of individual connections refugees make with people in the sector and the perception of affective care that mediated their interactions.

While loneliness and isolation are not specific to refugees, several aspects of the refugee experience emerged that underscore the importance of being able to access this type of emotional support, whether from migrant organisations or, as outlined below, from informal social contacts. As the quote below illustrates, the decision to leave one's home country to seek asylum elsewhere affects every aspect of refugees' future lives and makes the ability to access reliable and ongoing support crucial to a refugees' efforts to re-establish themselves in the new country context:

> *I think the most important thing is I see myself just beginning my life, second life after I moved to UK. It's a clean notebook that I'm just writing my own things. So, it is always good to communicate such people like [staff member at a refugee organisation] the people who help us throughout that time, up to now.* (Fazil, refugee participant)

Several participants confirmed that organisations had "always been there to help" and had "stood with them". This perception of their role was shared by practitioners. One staff member in a senior management role explained to new advisers during a discussion session that for her, the role of an integration service was precisely this, to "walk alongside refugees during this part of their journey". Doing this, and doing it well seemed fundamental to practitioners' engagement with their client group. As one participant explained, her family key worker in organisation 1 was somebody who did "her work really with love". This care was experienced too through a sense that caseworkers understood refugees' situations:

> *She's [caseworker] really kind and she does everything, like, with her heart, do you know what I mean? Because she knows refugees, she knows about all hard experiences and situations that all refugees were in.* (Miriam, refugee participant)

Others, like Elena, spoke of the warmth created by her contact with a partner organisation and linked this to becoming integrated into the city where she was now living:

> *They have help for you to try to maintain as a warm environment for you, to integrate you into the city.* (Elena, refugee participant)

Experiences of care, understanding and kindness then can influence refugees' feelings of belonging and acceptance (Yuval-Davis 2006) and in turn their perceptions of integration. While funding requirements for refugee sector organisations typically focus on outcomes such as employment and settled housing, refugees' own descriptions of the support they have received place equal value on these emotional dimensions of their connections with refugee sector organisations.

### 7.3. Resisting: Enabling Access to Rights

While care and connection featured highly in refugees' accounts of accessing refugee sector services, more practical concerns were also apparent. The demands of the asylum process and support systems in the UK require asylum route refugees to transition through a maze of complex systems at various points (Strang et al. 2017). These systems may be particularly impenetrable to those who are new to the country or lack confidence in English. Within these systems, the refugee-sector had had to step in to provide essentials, including access to food, money and clothing at numerous points. Frequently, this was most acute during the asylum process when people are excluded from mainstream welfare systems:

> *They [organisation 1] made sure the registration for my son's school and accommodation [ . . . ] And there is a form that I'm supposed to have entered for while in pregnancy, I didn't do that, so they helped me with that. [ . . . ] and they also supported me with clothing for my son and the baby as well . . . I needed to have an account for the clothing grants, I got a letter from [organisation 1] to open the accounts, though it wasn't easy, but with their support I was able to—[ . . . ] it's for their support that made it possible.* (Aisha, refugee participant)

Aisha went on to describe how a community organisation whose offices were next to those of the Home Office had been there to advise on the legal aspects of her case, knowing when to signpost her to her lawyer when she faced the threat of removal. Further, the Centre had delivered a children's programme and opportunities for her to socialise to counter the isolation during the asylum process, and had offered childcare when she was reporting to the Home Office, a requirement for every asylum seeker in the UK:

> *All through the process, they are always supportive. At a point, if I had to go to the Home Office [to sign in], I would drop my daughter in their office, and I would be free to go, and come back for the baby.* (Aisha, refugee participant)

Looking after Aisha's child was not an apolitical act of support however. The organisation intentionally offered this service to families as a strategy to avoid their being detained while reporting; the theory being that parents would not be detained if that risked their children being left alone in the UK. While Aisha's decision to leave her child points to the

gendered barriers experienced by individuals who are navigating these systems, the organisation's agreement to look after her child whilst she reported to the Home Office illustrates the ways in which acts of care can serve a more subversive purpose. This suggests the role of the refugee-sector goes beyond acts of care, to encompass acts of resistance to the hostile immigration landscape.

For some participants, provision of practical support and advice by the refugee sector had extended over a number of years and across multiple transitions in their lives, both personal and as regards their immigration status.

> *Even I have very difficult life with Home Office housing process but [partner organisation] was behind me, they help me . . . [ . . . ] they are still working [to help me] . . . after I get refugee leave to remain, they finish all the child benefit . . . from the reception to the manager, yes I've known them for seven years . . .* (Tsegaye, refugee participant)

Indeed, although refugees nominally experience improved access to rights once they have been recognised and granted leave to remain by the Home Office, these can be illusory in practice. Discussions with Integration Advisers, whose nominal role was to focus on integration goal setting and planning, revealed that addressing practical, basic needs often dominated the early stages of their work with people who were transitioning between the asylum support and mainstream welfare systems:

> *[There are] barriers we find in our work that prevent us focusing on work we'd like to do . . . a lot clients want to focus on financial stability and housing [ . . . ] It's difficult to focus on long-term objectives when you don't have stable income.* (Jenny, integration adviser)

The links between provision of practical support and improved feelings of acceptance show that it can be difficult if not impossible to untangle the emotional impact of support from its more practical manifestations:[4]

> *Thank goodness, I can eat, I have a roof over my head, and I have clothes, and they made me feel—they make you feel respected, like a living human being. Other than that, I don't really need much else.* (Mahmoud, refugee participant)

Practical support, it seems, can overlap with acts of care and also serve a subversive, political function. Practitioners from the refugee sector perceived "walk[ing] alongside refugees", as an active act of protecting refugees rights and, we argue, of resistance. Practitioners explained that they attempted to redress their clients' marginalisation through advice and support, something which occupied a great deal of their time:

> *Many statutory organisations don't understand the rights refugees have or their obligations to support them. In a landscape where the systems, structures and processes in this country are not clear for anyone, they are particularly unclear for marginalised populations, such as refugees. We spend most of our time firefighting with inaccessible systems.* (Maya, Integration Adviser)

Indeed, for practitioners the time they expended in trying to overcome systems barriers often detracted from supporting clients to work towards what funders and governments typically situate as desirable integration goals, including employment and education. This, some practitioners noted, was only going to become ever more the case in future as "integrating into austerity" became the reality of their beneficiaries' lives, and their own working environment.

## 8. Discussion

Our study goes some way to addressing the gap in research capturing refugees' and practitioners' perspectives of the practical and emotional role that refugee sector organisations, situated as both a social connection in their own right and a node in a wider network, play in refugees' pathways to social and economic inclusion. Firstly, refugees highlighted the role the sector played in both connecting them to grassroots civil society organisations and to informal peer networks. As has been previously argued, this in turn can enable refugees to build a foundation for more tangible outcomes, including

employment, by acting as a node linking them to wider networks and opportunities (Calò et al. 2021). Service providers can also play a key role in helping refugees to build social capital through connecting them with their co-ethnic and diverse communities (Pittaway et al. 2016). One social connection leading on to another, over time and across people's changing needs and circumstances is an ideal illustration of the ways in which connections sit upon a continuum and, like integration itself, are better conceptualised as processual rather than static (Strang and Quinn 2021).

Second, it highlights the equal value refugees place on the affective role the sector plays, expressed as a human connection and compared to the love and kindness of family and friends. This consistent support provided over time was perceived as important precisely because it extended beyond a functional, transactional role and was experienced as acts of care. Refugees felt they had created personal ties and coalitions with service practitioners which had contributed to feelings of welcome, self-worth and a sense of belonging (Yuval-Davis 2006).

While refugees' accounts speak to the importance of the care they experience from the refugee sector, practitioners' accounts of their work illustrate some of the tensions and potential pitfalls in performing acts of care. The asymmetry inherent in the acts of care described by Darling finds expression too in the current study (Darling 2011); while several refugee participants felt that their relationships with practitioners were akin to close familial or friendship ties, practitioners themselves did not speak in these terms. There was also little evidence of opportunities for refugees to reciprocate with their own experiences and skills, outside organisation 1's Peer Project.

One way in which refugee sector organisations addressed this was through their engagement, with and on behalf of beneficiaries, in daily acts that sought to overcome and subvert statutory systems barriers that face recently recognised refugees. In these acts of what we have termed *"practice-based resistance"*[5]—challenging housing decisions, contacting jobcentres, advising on rights in relation to work and schooling—practitioners were able to, in the words of one of the service providers, "walk alongside refugees" not as caregivers, but as more equal partners. In this way, refugee sector organisations demonstrated their ability to move beyond charitable care to engage in work that recognises and challenges the structural constraints that shape refugees' lives. This, we suggest, is one of the key roles of refugee sector organisations and is a role that is ever more important given the degradation of the external environment in which they operate. Practically, this work can be seen as a facilitator of integration, acting as a bulldozer that moves barriers from along the way. Less tangibly, refugees' accounts indicate that having someone alongside them can itself be a key part of starting to feel integrated—expressed variously through feeling settled, feeling belonging or feeling at home. This is particularly the case where this support is offered and received even before the government formalizes their refugee status through granting permission to stay.

As such our findings demonstrate the increasingly salient role of refugee-sector organisations in countering refugees' marginalisation in society through acts of care as resistance. The embedded humanity in service provision works as a strategy of resistance to counter the hostile environment to immigration which has systematically eroded refugees and asylum seekers' rights and access to services over the last decade. This "welfare restrictionism" (Phillimore 2015, p. 578) not only imposes new vulnerabilities through structural inequalities. It also perpetuates social disconnect which undermines pathways to belonging. In this context, refugee-sector service provision faces the challenge of not only addressing gaps in essential welfare provision but promoting a vision of integration that foregrounds refugees' aspirations and emphasises the importance of belonging and well-being in refugees' experiences of making their lives in Britain.

Nonetheless, caution is needed to avoid over-romanticising the power of the refugee-sector. While the refugee-sector is central in enabling people subject to immigration control to survive the hostile environment (Saunders and Al-Om 2022), contextualised analysis of refugees and service providers' narratives of engagement highlights that practice-based

resistance is not only necessitated, but also constrained by the hostile immigration regime and neoliberal welfare provision. Amidst increased demands and constrained resources, migrant organisations are increasingly left to grapple with the tensions inherent in pursuing human right based approaches to service delivery, whilst also being compelled to fill gaps in statutory provision. Crucially, the hostile environment does not exist in isolation, but has been intertwined with wider welfare erosion and privatisation; as argued by Mesarič and Vacchelli: "[a]usterity politics produce vulnerability while at the same time restricting mechanisms set up to address it" (Mesarič and Vacchelli 2021, p. 3101). As illustrated by our findings, service providers are often left firefighting access to essential rights and resources which take precedence over other areas of integration. Nonetheless, as highlighted by refugees themselves, the practice-based resistance experienced as care is in of itself central to the process of regaining confidence and rebuilding networks which enable refugees to constitute themselves as autonomous actors.

## 9. Conclusions

This paper has explored refugees' experiences of refugee-sector integration advocacy and support as well as the views of the practitioners who deliver them, with a focus on embodied meanings of care within an otherwise hostile immigration regime. Our findings contribute to further understanding of refugee-sector services as providers of welfare. Crucially, the external context characterised by ever-increased entanglements of immigration policy and welfare restrictionism (Phillimore 2015) underscores the importance of multidimensional advocacy and support for refugees to navigate entitlements which they can rarely readily access. In focusing on meanings refugees attach to service provision, our paper illustrates that feeling integrated does not solely depend on achieving tangible outcomes such as paid employment, but can be shaped by the care provided by refugee sector organisations which furthers refugees' feelings of inclusion and belonging.

In exploring refugees' lived experiences of this care, we have highlighted the importance of human connection which is embedded in the ethos of the refugee sector. This is not only about welcoming refugees, but in of itself acts as an enabler for refugees to develop confidence and further networks of support which are central to their pathways to inclusion. In adopting a practice-research engagement design (Brown et al. 2003) which has engaged both beneficiaries and service providers, we have been able to illuminate the politicisation of care through the strategic role of "*practice-based resistance*" as a means of challenging the very architecture of the hostile environment which has been built around deterrence, lack of choice and dehumanisation of refugees and asylum seekers. Nonetheless, while practice-based resistance exists to resist the hostile environment, it is also increasingly constrained by it. Under austerity measures, the (formalised and grassroots) refugee sector will increasingly struggle to deliver care above and beyond basic welfare provision as it remains chronically under-resourced. This paper calls for sustainable funding in recognition of the important role the refugee sector plays in exercising practice-based resistance and, by extension, supporting integration.

**Author Contributions:** Writing—Original Draft Preparation, E.K., H.B., L.K. and M.V.-E.; Conceptualization, E.K., H.B., L.K. and M.V.-E.; Methodology, E.K., H.B., L.K. and M.V.-E.; Formal Analysis, E.K., H.B. and L.K.; Investigation, E.K., H.B., L.K. and M.V.-E.; Writing—Review & Editing, E.K., H.B., L.K. and M.V.-E.; Project Lead, M.V.-E. All authors have read and agreed to the published version of the manuscript.

**Funding:** This research was funded by The Asylum, Migration and Integration Fund (AMIF), grant number UK/2020/PR/0104.

**Institutional Review Board Statement:** The study was conducted in accordance with the Declaration of Helsinki, and was reviewed and approved by the Queen Margaret University Ethics of Research Committee (ref: REP 0244).

**Informed Consent Statement:** Informed consent was obtained from all participants involved in the study.

**Data Availability Statement:** The data presented in this study is not publicly available due to confidentiality obligations towards the study participants. No names have been used in the paper to respect these obligations. The data presented in this paper is part of a larger study, which other outputs are still under development.

**Acknowledgments:** The authors would like all the Integration Service partnership organisations, staff and beneficiaries for their contributions to the project. We would also like to thank Arek Dakessian for his contribution to research planning and data collection.

**Conflicts of Interest:** The authors declare no conflict of interest.

## Notes

1    This work was undertaken as part of the AMIF-funded 'New Scots Integration—A Pathway to Social and Economic Inclusion' ABM3 Project (UK/2020/PR/0104).

2    https://www.whitehorsedc.gov.uk/wp-content/uploads/sites/3/2022/04/Minister-Announcement-letter-to-Local-Authorities.pdf (accessed on 30 September 2022).

3    https://togetherwithrefugees.org.uk/ (accessed on 30 September 2022).

4    Preliminary findings and initial ideas around the practical and emotional functions and perceptions of care were presented at the International Conference 'Migration and Care', in June 2022.

5    We draw here on ideas around 'resistance' as practices relevant to understand actors and processes of integration within a context of bordering practices (Vera Espinoza 2022), as well as drawing from literature on migrant solidarity (Ataç et al. 2016; Vickers 2016).

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
