# Peer review of "From Acts of Care to Practice-Based Resistance: Refugee-Sector Service Provision and Its Impact(s) on Integration"

_socsci, doi:10.3390/socsci12010039_

Round 1

Reviewer 1 Report

This manuscript, “From acts of care to practice-based resistance: the role and impact of refugee-sector service provision in the UK” deals with what is an important global issue, namely immigration.  Immigrants – defined as persons moving from one country to another – come in many forms, e.g., refugees, asylum seekers, illegal aliens, human trafficking victims, etc.  All so-called destination countries, to the best of my knowledge, have some restrictions, some laws and regulations, with respect to such persons.  Countries do not, in other words, simply have “open borders.”  The laws and regulations govern numbers and policies and procedures established by the authorities. The authors of this paper obviously disagree with the immigration governing mechanisms in the UK.  Given their disagreement, what they have produced here is an advocacy paper supporting what they term “practice-based resistance” to these mechanisms, which they say have produced a “hostile environment” to immigration.  As advocates, the authors have an obvious bias in favor of immigrants and against the policies.  This would be fine if this were intended simply as an opinion piece, but it is not fine for a work of research and scholarship intended to enlighten its readers.  As a consequence of the authors’ approach, we do not learn anything about just what are the policies and procedures that constitute this “hostile environment,” nor do we learn from anyone what the justification for those policies and procedures might be.  We thus have no context for the discussion.  Putting aside its biased nature, the data collection itself is very limited – a small number of persons and a few focus groups – all of whom seem to represent the one point of view.  On top of that, the charts included to show some quantitative findings are not clear to me.  In sum, although it addresses an important topic, I recommend rejection of this paper.  

Author Response

Response to Reviewer 1

Feedback: This manuscript, “From acts of care to practice-based resistance: the role and impact of refugee-sector service provision in the UK” deals with what is an important global issue, namely immigration.  Immigrants – defined as persons moving from one country to another – come in many forms, e.g., refugees, asylum seekers, illegal aliens, human trafficking victims, etc.  All so-called destination countries, to the best of my knowledge, have some restrictions, some laws and regulations, with respect to such persons.  Countries do not, in other words, simply have “open borders.”  The laws and regulations govern numbers and policies and procedures established by the authorities. The authors of this paper obviously disagree with the immigration governing mechanisms in the UK.  Given their disagreement, what they have produced here is an advocacy paper supporting what they term “practice-based resistance” to these mechanisms, which they say have produced a “hostile environment” to immigration.  As advocates, the authors have an obvious bias in favor of immigrants and against the policies.  This would be fine if this were intended simply as an opinion piece, but it is not fine for a work of research and scholarship intended to enlighten its readers.  As a consequence of the authors’ approach, we do not learn anything about just what are the policies and procedures that constitute this “hostile environment,” nor do we learn from anyone what the justification for those policies and procedures might be.  We thus have no context for the discussion.  Putting aside its biased nature, the data collection itself is very limited – a small number of persons and a few focus groups – all of whom seem to represent the one point of view.  On top of that, the charts included to show some quantitative findings are not clear to me.  In sum, although it addresses an important topic, I recommend rejection of this paper.  

Our response: We would like to thank the reviewer for reading our manuscript, although we would have welcomed a more serious engagement with the text. We have listed below our responses and description of the changes made to the new version of the manuscript to address each individual point made by the reviewer. The reviewer’s overarching opposition to our paper seems to be based on “bias” in favour of immigrants which has led them to recommend rejection of our paper. We strongly oppose this assessment of our paper. Notably, no part of our original manuscript, or its revised version, advocates for “open borders” or total lack of immigration controls, as argued by the reviewer. In fact, most of the data presented in this paper comes from recognised refugees to whom the UK law grants full access to social security, social housing, labour market and other key institutions of the welfare state. These dimensions are central to refugee integration, which, as described by our paper, is also an area of policy promoted by the UK Government. While we agree with the reviewer’s assessment that the authors disagree with the immigration governing mechanisms in the UK, we do not agree that our paper constitutes a mere “opinion piece”; the findings presented in this paper evidence the harmful consequences of the UK policy environment, particularly highlighting its role in contributing to the systematic structural barriers experienced by recognised refugees who have a legal right to these services and rights. In addition to the evidence presented in our paper, the UK policy approach to forced migration has been widely condemned by leading national and international human rights organisations including the UNHCR, and further challenged by wealth of scholarly evidence spanning over decades highlighting these policies as a barrier to integration and social cohesion, and as a causal factor in fuelling destitution, intersectional vulnerabilities and negative consequences to mental and physical wellbeing and individual and community safety. Some of this key evidence has now been further emphasised in our revised manuscript. We have also taken the reviewer’s feedback by adding an in-depth explanation about the hostile environment (which has been the UK Government’s official policy stance, rather than our own term), the policies which have been implemented to enforce it, their consequences and evidence problematising the effectiveness of this policy approach to achieve its aims (to reduce immigration to the UK).

Responses to individual points made by reviewer 1:

As a consequence of the authors’ approach, we do not learn anything about just what are the policies and procedures that constitute this “hostile environment,” nor do we learn from anyone what the justification for those policies and procedures might be. We thus have no context for the discussion.    

  • We have now added a section introducing the UK official hostile environment policy, some examples of what it consists of, its justifications as stated by the then government, with cited evidence of its consequences, concluding with a point problematising the effectiveness of the policy to achieve its purpose. Further links to the hostile environment have also been added in the discussion and conclusion.

Putting aside its biased nature, the data collection itself is very limited – a small number of persons and a few focus groups – all of whom seem to represent the one point of view. On top of that, the charts included to show some quantitative findings are not clear to me.  In sum, although it addresses an important topic, I recommend rejection of this paper.  

  • We respectfully disagree with the point the reviewer raises about our sample size which does not seem to be informed by qualitative research scholarship. Our article is based on data from 30 refugees and 20 practitioners, which is well within the sample range recommended for qualitative interview research (Dworkin, 2012). In fact, many well cited papers highlight the importance saturation over sample size in qualitative research (Sandelowski, 1995; Boddy, 2016). The key themes, evident in our both quantitative and qualitative findings, strongly indicate a saturation was reached during the data collection. The quantitative findings are limited to counts which cannot be used to establish causal effects or enable the use more nuanced statistical techniques. As explained in our paper, this was used primarily to inform the focus of our qualitative analysis on refugee-sector organisations, rather than to make generalisations about the wider refugee population in Scotland which would have required a larger sample size and a different method of recruitment. We have now added further explanation on the methods and made the figures clearer.

Reviewer 2 Report

This is an interesting and well written paper with potential to make a contribution to the fields of refugee and/or third sector studies. The authors have collected some interesting empirical materials and use verbatim well to illustrate the views and experiences of refugees and service providers. There are some over-generalisations, for example the focus on 3rd sector is not really a feature of Big Society but pre-dated Big Society and has always been core to ASR services. See literature of on the mixed economy of welfare. However, while the 3rd sector has always had a role in service provision the emphasis changed during austerity. Note that in 1980s to 2000s RCOs were the only organisations with the specialist knowledge needed to support new refugees and asylum seekers – see Zetter et al.

The introduction section places the paper in context but it would be good to see here a clear statement of the question being addressed in the paper and what the contribution to knowledge is in theoretical/methodological and empirical terms. Here focus on what you are adding to integration theory or third sector studies (or both).

The literature review section is rather light. It is important that you cover the state of knowledge in this field and outline the key theoretical or conceptual contributions intended, offering some kind of theoretical or conceptual framework. For example there is a huge body of knowledge on integration yet while you refer to integration repeatedly you do not really state what you mean by integration and integration processes. This would be your opportunity to identify gaps in that body of knowledge. The Indicators of Integration - to what extent do these account for care? Could care be considered under the aegis of "safety" or "stability" or does it relate to "shared responsibility"? Also there is a huge body of knowledge on care - what is the state of the art here and how might it be connected with thinking on integration processes? It is important to link these in a narrative that leads logically to your key questions. Also in the methods and findings ideas around social connections are widely invoked. There is a massive literature on this which is not considered at all - at the very least you need to consider the ways that social connection have been utilised in thinking around integration.

Minor points but the IoI were produced by and for the Home Office not backed by and asylum seekers in England and Wales are entitled to secondary health care. Given the international readership of this journal you need at some point to describe the UK policy position re asylum seekers and refugees - how refugees come to be in Scotland, the absence of a UK wide integration policy, dispersal and notice periods. Are you talking about resettlement or Convention refugees? For the former the institutional framework is very different.

In methods more data is needed about your sample - how was it collected, how were people chosen, did they self-select? Who was included in your sample? Who was not? What about gender, age, religion? And then why did you only use a sub-sample for this paper - how was that identified? You use multiple methods - how do these complement each other/triangulate - what does each method add? How was your data analysed? Include a bullseye map as an example figure so readers can see what you are talking about.

The findings were interesting and the figures useful but please check that your categories are mutually exclusive and for Figure 1 place the bars in order of the frequency of responses (or reverse order) - it makes the chart easier to read. You state in findings "in the content analysis" - not sure what you mean here. You also say your findings are biased because of sampling size and technique - biased in what way? What can you assert on the basis of your actual sample? What can you not argue? Any gender differences?

Discussion needs to be connected to the wider literature. Ideas of practice based resistance are interesting but perhaps develop this idea using existing work on resistance or explore these resistances as a micropolitical strategy for organisations which lack any power to influence policy directly. You make a couple of statements alluding to comparison of NGO services with that of the state - but you cannot make such a comparison unless you explore state provision. So if these claims have been made by respondents make that clear to the reader.

There is a lot going on in this paper - integration, care, emotions, resistance and third sector actions. To make the paper coherent and decide on your main contributions you need to drill deeper into existing work and then in the discussion show exactly how your findings add to knowledge.

Author Response

Response to Reviewer 2

Feedback: This is an interesting and well written paper with potential to make a contribution to the fields of refugee and/or third sector studies. The authors have collected some interesting empirical materials and use verbatim well to illustrate the views and experiences of refugees and service providers. There are some over-generalisations, for example the focus on 3rd sector is not really a feature of Big Society but pre-dated Big Society and has always been core to ASR services. See literature of on the mixed economy of welfare. However, while the 3rd sector has always had a role in service provision the emphasis changed during austerity. Note that in 1980s to 2000s RCOs were the only organisations with the specialist knowledge needed to support new refugees and asylum seekers – see Zetter et al. The introduction section places the paper in context but it would be good to see here a clear statement of the question being addressed in the paper and what the contribution to knowledge is in theoretical/methodological and empirical terms. Here focus on what you are adding to integration theory or third sector studies (or both). The literature review section is rather light. It is important that you cover the state of knowledge in this field and outline the key theoretical or conceptual contributions intended, offering some kind of theoretical or conceptual framework. For example there is a huge body of knowledge on integration yet while you refer to integration repeatedly you do not really state what you mean by integration and integration processes. This would be your opportunity to identify gaps in that body of knowledge. The Indicators of Integration - to what extent do these account for care? Could care be considered under the aegis of "safety" or "stability" or does it relate to "shared responsibility"? Also there is a huge body of knowledge on care - what is the state of the art here and how might it be connected with thinking on integration processes? It is important to link these in a narrative that leads logically to your key questions. Also in the methods and findings ideas around social connections are widely invoked. There is a massive literature on this which is not considered at all - at the very least you need to consider the ways that social connection have been utilised in thinking around integration. Minor points but the IoI were produced by and for the Home Office not backed by and asylum seekers in England and Wales are entitled to secondary health care. Given the international readership of this journal you need at some point to describe the UK policy position re asylum seekers and refugees - how refugees come to be in Scotland, the absence of a UK wide integration policy, dispersal and notice periods. Are you talking about resettlement or Convention refugees? For the former the institutional framework is very different. In methods more data is needed about your sample - how was it collected, how were people chosen, did they self-select? Who was included in your sample? Who was not? What about gender, age, religion? And then why did you only use a sub-sample for this paper - how was that identified? You use multiple methods - how do these complement each other/triangulate - what does each method add? How was your data analysed? Include a bullseye map as an example figure so readers can see what you are talking about. The findings were interesting and the figures useful but please check that your categories are mutually exclusive and for Figure 1 place the bars in order of the frequency of responses (or reverse order) - it makes the chart easier to read. You state in findings "in the content analysis" - not sure what you mean here. You also say your findings are biased because of sampling size and technique - biased in what way? What can you assert on the basis of your actual sample? What can you not argue? Any gender differences? Discussion needs to be connected to the wider literature. Ideas of practice based resistance are interesting but perhaps develop this idea using existing work on resistance or explore these resistances as a micropolitical strategy for organisations which lack any power to influence policy directly. You make a couple of statements alluding to comparison of NGO services with that of the state - but you cannot make such a comparison unless you explore state provision. So if these claims have been made by respondents make that clear to the reader. There is a lot going on in this paper - integration, care, emotions, resistance and third sector actions. To make the paper coherent and decide on your main contributions you need to drill deeper into existing work and then in the discussion show exactly how your findings add to knowledge.

Our response: We would like to thank the reviewer for their insightful and constructive feedback. We have listed below our responses and description of the changes made to the new version of the manuscript to address each individual point made by the reviewer. In particular, the new version of the manuscript: more closely contextualises our findings and discussion in relation to wider scholarship on social connections in refugee integration, with a clearer statement on our contribution; includes a more in-depth discussion on conceptualisations of care in relevance to our findings; and explains in more detail our methodology, with clearer presentation of figures. We hope that this revised version sufficiently addresses the suggestions made by the reviewer.

Responses to individual points made by reviewer 2:

There are some over-generalisations, for example the focus on 3rd sector is not really a feature of Big Society but pre-dated Big Society and has always been core to ASR services. See literature of on the mixed economy of welfare. However, while the 3rd sector has always had a role in service provision the emphasis changed during austerity. Note that in 1980s to 2000s RCOs were the only organisations with the specialist knowledge needed to support new refugees and asylum seekers – see Zetter et al.

  • Part of this feedback was addressed in our original submission, which already stated that “Charities have arguably played an important role in addressing social needs even prior to the emergence of the contemporary British welfare state”, citing a chapter from established scholars working on the area of history of mixed economy of welfare in Britain (Harris and Bridgen 2012). We have now removed the argument about Big Society due to lack of scope to explore the nuances related to the role of third sector in this. In a different part of our original submission we stated that “refugee integration in the UK has been spearheaded by a strong third sector.” While the work of Zetter and colleagues focuses on refugee community organisations which our research has not directly engaged with, we have taken this feedback on board by adding a sentence to further highlight the central role of these organisations in the UK policy context. We have also cited the work of Zetter to address the other reviewer’s earlier comment about the hostile environment.

The introduction section places the paper in context but it would be good to see here a clear statement of the question being addressed in the paper and what the contribution to knowledge is in theoretical/methodological and empirical terms. Here focus on what you are adding to integration theory or third sector studies (or both).

  • We have now inserted, at the end of the introduction section, a clearer statement of our contribution to both integration and third sector studies, namely an empirical contribution by means of inclusion of little-heard insights from integration service practitioners; and an opportunity to expand our conceptual understandings of the role of the refugee sector and its impact upon integration, focusing on the role of this type of social connection in functional and emotional aspects of integration.

The literature review section is rather light. It is important that you cover the state of knowledge in this field and outline the key theoretical or conceptual contributions intended, offering some kind of theoretical or conceptual framework. For example there is a huge body of knowledge on integration yet while you refer to integration repeatedly you do not really state what you mean by integration and integration processes. This would be your opportunity to identify gaps in that body of knowledge. The Indicators of Integration - to what extent do these account for care? Could care be considered under the aegis of "safety" or "stability" or does it relate to "shared responsibility"? Also there is a huge body of knowledge on care - what is the state of the art here and how might it be connected with thinking on integration processes? It is important to link these in a narrative that leads logically to your key questions.

  • We have now included a section that summarises existing literature and gaps in knowledge around the role of social connections in integration, and highlights too some of the more pertinent debates around the meaning(s) ascribed to integration. In our discussion and conclusion, we have now more explicitly drawn links with existing debates around integration and suggest that both care, and the practice-based resistance that emerged in our findings are elements of refugee sector provision that facilitate integration and contribute not only to functional outcomes such as access to services and rights, but also to integration’s more affective and emotional elements. We have now added a longer discussion about care to the conceptual section. On one hand, this section now acknowledges the wide range of literature on the understandings of care. On the other hand, it shows our focus on the relational aspect of care, emphasising the relevance of connections, values and practice, which are core to our argument and links to our relational approach to integration. We also explored the affective connotations of care as a value and the possibilities of the politicisation of care as resistance.

Also in the methods and findings ideas around social connections are widely invoked. There is a massive literature on this which is not considered at all - at the very least you need to consider the ways that social connection have been utilised in thinking around integration.

  • As above, we have now inserted a section which summarises key literature around the role of social connections in integration, highlighting some of the ways in which this paper can deepen understandings of their role in integration processes.

Minor points but the IoI were produced by and for the Home Office not backed by and asylum seekers in England and Wales are entitled to secondary health care. Given the international readership of this journal you need at some point to describe the UK policy position re asylum seekers and refugees - how refugees come to be in Scotland, the absence of a UK wide integration policy, dispersal and notice periods. Are you talking about resettlement or Convention refugees? For the former the institutional framework is very different.

  • We have now added a citation for the point about healthcare entitlements (BMA, 2022). We have also added a sentence on the policy context section explaining the historical position of Glasgow as a dispersal city, with a mention of other routes that are beyond the scope of our paper. We have also clarified in the methodology section that our participant sample consists fully of asylum-route refugees. While our paper notes that the HO endorses IoI, we have opted not to phrase this a “being produced by” as the authors of the original version, and many of the authors of the third iteration of the framework were commissioned academics (whose other work has often also been critical of HO), as opposed to HO employees.

In methods more data is needed about your sample - how was it collected, how were people chosen, did they self-select? Who was included in your sample? Who was not? What about gender, age, religion? And then why did you only use a sub-sample for this paper - how was that identified? You use multiple methods - how do these complement each other/triangulate - what does each method add? How was your data analysed? Include a bullseye map as an example figure so readers can see what you are talking about.

  • A paragraph has now been added in the methodology section giving more detail on how we purposively sampled from the current beneficiary cohort to to reflect a range of experiences and perspectives according to gender, nationality and language proficiency. It also explains why the timing of this paper led us to exclude a small number of interviews from our sample of data for this paper. A sentence has been added to explain how the quantitative and qualitative methods complement one another. We have also edited our figures, and added a figure of the bullseye map.

The findings were interesting and the figures useful but please check that your categories are mutually exclusive and for Figure 1 place the bars in order of the frequency of responses (or reverse order) - it makes the chart easier to read. You state in findings "in the content analysis" - not sure what you mean here. You also say your findings are biased because of sampling size and technique - biased in what way? What can you assert on the basis of your actual sample? What can you not argue? Any gender differences?

  • We have now redone the figures to make them clearer. We have added further explanation on how we counted the bullseye connections and a figure illustrating the bullseye to strengthen this section. We have also clarified that as the connections were recorded on the maps per group as opposed to per participant, demographic comparisons were not made at this stage. As outlined in this section, the purpose of these counts was to inform the direction of subsequent qualitative analysis of the transcripts, as presented in the following findings sections. We have now removed the mention of bias, instead clarifying that, as our participants were recruited solely through our partner organisations, this may explain why these partners featured as the most important connections in both figures (which may have not been the case had we used other recruitment approaches).

Discussion needs to be connected to the wider literature. Ideas of practice based resistance are interesting but perhaps develop this idea using existing work on resistance or explore these resistances as a micropolitical strategy for organisations which lack any power to influence policy directly. You make a couple of statements alluding to comparison of NGO services with that of the state - but you cannot make such a comparison unless you explore state provision. So if these claims have been made by respondents make that clear to the reader.

  • We have now included further discussion on resistance, linked to the notion of care that the paper mobilises. We have also made more explicit connections to this analysis in the presentation and discussion of the findings, in order to make it clear. Our comparisons to statutory services are limited to our participants experiences of the structural barriers they have experienced in accessing key rights and services. While the revised version explores our conceptualisations of resistance in more detail, we don’t agree with the presumption that the organisations would lack any power to influence policy directly – whilst our research has only engaged with practitioners delivering integration services, some of them work in organisations which have their own policy department, and which actively engage in policy influencing through various means, much like many other established refugee-sector organisations in the UK.

There is a lot going on in this paper - integration, care, emotions, resistance and third sector actions. To make the paper coherent and decide on your main contributions you need to drill deeper into existing work and then in the discussion show exactly how your findings add to knowledge.

  • We have re-shaped and re-formulated our literature sections to deepen our engagement with existing work, and to embed a clearer overall narrative that focuses on the three ways in which refugee sector organisations contribute to integration: through connecting, caring and resisting. This in turn, we suggest, highlights elements of integration that conceptually and empirically remain under-explored.

Reviewer 3 Report

The paper ‘From acts of care to practice-based resistance: the role and impact of refugee-sector service provision in the UK’, provides an interesting take on the UK asylum/refugee sector from the perspective of both people running third sector initiatives and those using them. There is some rich data presented in the findings sections and there is potential to do much more with this. At the moment, however, the paper is rather thin in terms of its theoretical contribution. The main debates/literature/concepts that the paper is engaging with are not adequately identified and this follows through to the findings – these are not linked back to the debates/existing literature as they weren’t sketched out early on. To improve this paper, the conceptual/theoretical contribution will need to be made much clearer and then the findings will need to be brought into dialogue with these debates more effectively.

My more specific observations/suggestions continue below:

1)      Integration seems to be a key focus of this paper but it appears rather intermittently as it stands. The paper requires a much deeper theoretical engagement with debates about integration (in relation to people seeking asylum/refugees) and the role of third sector organisations in these processes. If this is, indeed, the main concept you want to make reference to you might consider having a lengthy section on it in the literature review to join the dots between existing work and your major contribution. At the moment, it feels like you’re somewhere between integration and care as the main contributions although the former seemed to be more evident throughout the paper – the latter takes up a very short paragraph in the literature review section. So, some decisions to be made here about stating your main contribution in relation to debates about integration and engaging with the academic debates that relate to the concept. What is your work adding to that? I think it is but needs to be reiterated throughout and given focused attention in the literature review.

2)      The section on UK policy context required some more info. It would be useful for readers to learn about the Hostile Environment, for example, and the broader climate vis-à-vis immigration in the UK. This isn’t a huge addition but would be useful for those not familiar with the UK context and changing debates re. immigration and policies on refugees and people seeking asylum.

3)      As a reader, I found the rich qualitative quotes more informative and useful when compared to the bar graphs. I would consider removing these as I’m not sure they are adding quite as much to the paper. A more minor issue: Does AMIF need more careful introduction and explanation?

4)      On page 8 you mention that your findings could be considered biased. I would avoid this language. You have used a qualitative methodology for good reason and as you say, you can’t make sweeping generalisations but certain patterns can be identified. This is the nature of qualitative research so I wouldn’t concede quite so much here. Perhaps you could talk about your approach to sampling a little more carefully instead.

5)      As stated above, the quotes within the findings sections were really interesting and rich. It was great to see an attempt to bring the accounts of service providers/staff members and participants together. However, at the moment these insights are rather isolated and not brought back into dialogue with existing debates about integration in the asylum/immigration literature.

6)      I thought the paper could be more critical about the reasons for this level of support/care being provided by third sector organisations. Does this reflect a failing of the state and state policies (or just, quite simply, its absence) in relation to integration programmes? So, a bit more critical commentary on the broader political, economic conditions that have got us to this situation would be really good to see.

There are some more critical arguments introduced in the discussion section about the functioning of third sector organisations but it would be good to see these flagged earlier in the paper so they’ve been set up ahead of the finale. Some of these felt a little ‘dropped in’ rather late in the day instead of being worked through the entire paper.

7)      Finally, you state that the paper goes some way to addressing the gap in research, but to what end. This is your opportunity to push why this is important. You filled the gap and what has that enabled – what has that enabled us to see better/understand?

Many thanks for giving me the opportunity to engage with your work. I enjoyed reading it. There is potential here but as it stands the academic thread running through your paper i.e. the central argument and theoretical contributions, as well as the main concepts, need to be more clearly outlined and brought into dialogue with your research. This will allow you to make the most of your rich empirical findings and state very clear contributions to debates in the literature regarding integration, the role of third sector organisations and the experiences of people seeking asylum.

Author Response

Response to Reviewer 3

Feedback: The paper ‘From acts of care to practice-based resistance: the role and impact of refugee-sector service provision in the UK’, provides an interesting take on the UK asylum/refugee sector from the perspective of both people running third sector initiatives and those using them. There is some rich data presented in the findings sections and there is potential to do much more with this. At the moment, however, the paper is rather thin in terms of its theoretical contribution. The main debates/literature/concepts that the paper is engaging with are not adequately identified and this follows through to the findings – these are not linked back to the debates/existing literature as they weren’t sketched out early on. To improve this paper, the conceptual/theoretical contribution will need to be made much clearer and then the findings will need to be brought into dialogue with these debates more effectively.

My more specific observations/suggestions continue below:

1)      Integration seems to be a key focus of this paper but it appears rather intermittently as it stands. The paper requires a much deeper theoretical engagement with debates about integration (in relation to people seeking asylum/refugees) and the role of third sector organisations in these processes. If this is, indeed, the main concept you want to make reference to you might consider having a lengthy section on it in the literature review to join the dots between existing work and your major contribution. At the moment, it feels like you’re somewhere between integration and care as the main contributions although the former seemed to be more evident throughout the paper – the latter takes up a very short paragraph in the literature review section. So, some decisions to be made here about stating your main contribution in relation to debates about integration and engaging with the academic debates that relate to the concept. What is your work adding to that? I think it is but needs to be reiterated throughout and given focused attention in the literature review.

2)      The section on UK policy context required some more info. It would be useful for readers to learn about the Hostile Environment, for example, and the broader climate vis-à-vis immigration in the UK. This isn’t a huge addition but would be useful for those not familiar with the UK context and changing debates re. immigration and policies on refugees and people seeking asylum.

3)      As a reader, I found the rich qualitative quotes more informative and useful when compared to the bar graphs. I would consider removing these as I’m not sure they are adding quite as much to the paper. A more minor issue: Does AMIF need more careful introduction and explanation?

4)      On page 8 you mention that your findings could be considered biased. I would avoid this language. You have used a qualitative methodology for good reason and as you say, you can’t make sweeping generalisations but certain patterns can be identified. This is the nature of qualitative research so I wouldn’t concede quite so much here. Perhaps you could talk about your approach to sampling a little more carefully instead.

5)      As stated above, the quotes within the findings sections were really interesting and rich. It was great to see an attempt to bring the accounts of service providers/staff members and participants together. However, at the moment these insights are rather isolated and not brought back into dialogue with existing debates about integration in the asylum/immigration literature.

6)      I thought the paper could be more critical about the reasons for this level of support/care being provided by third sector organisations. Does this reflect a failing of the state and state policies (or just, quite simply, its absence) in relation to integration programmes? So, a bit more critical commentary on the broader political, economic conditions that have got us to this situation would be really good to see.

There are some more critical arguments introduced in the discussion section about the functioning of third sector organisations but it would be good to see these flagged earlier in the paper so they’ve been set up ahead of the finale. Some of these felt a little ‘dropped in’ rather late in the day instead of being worked through the entire paper.

7)      Finally, you state that the paper goes some way to addressing the gap in research, but to what end. This is your opportunity to push why this is important. You filled the gap and what has that enabled – what has that enabled us to see better/understand?

Many thanks for giving me the opportunity to engage with your work. I enjoyed reading it. There is potential here but as it stands the academic thread running through your paper i.e. the central argument and theoretical contributions, as well as the main concepts, need to be more clearly outlined and brought into dialogue with your research. This will allow you to make the most of your rich empirical findings and state very clear contributions to debates in the literature regarding integration, the role of third sector organisations and the experiences of people seeking asylum.

Our response: We would like to thank the reviewer for their encouraging and constructive feedback. We have listed below our responses and description of the changes made to the new version of the manuscript to address each individual point made by the reviewer. Our revised manuscript now more fully explains the UK policy context, provides a deeper discussion on social connections in integration and the theoretical conceptualisations of care which inform our paper. We have also more clearly articulated our contributions to existing literature. We hope these revisions satisfy the revisions requested by the reviewer.

Responses to individual points made by reviewer 3:

There is some rich data presented in the findings sections and there is potential to do much more with this. At the moment, however, the paper is rather thin in terms of its theoretical contribution. The main debates/literature/concepts that the paper is engaging with are not adequately identified and this follows through to the findings – these are not linked back to the debates/existing literature as they weren’t sketched out early on. To improve this paper, the conceptual/theoretical contribution will need to be made much clearer and then the findings will need to be brought into dialogue with these debates more effectively.

  • We have now brought in a new section in our literature review that highlights existing work around social connections and integration and have sought to further embed this throughout the paper. We have sought to clarify and consolidate our empirical and conceptual contributions throughout the paper.

Integration seems to be a key focus of this paper but it appears rather intermittently as it stands. The paper requires a much deeper theoretical engagement with debates about integration (in relation to people seeking asylum/refugees) and the role of third sector organisations in these processes. If this is, indeed, the main concept you want to make reference to you might consider having a lengthy section on it in the literature review to join the dots between existing work and your major contribution. At the moment, it feels like you’re somewhere between integration and care as the main contributions although the former seemed to be more evident throughout the paper – the latter takes up a very short paragraph in the literature review section. So, some decisions to be made here about stating your main contribution in relation to debates about integration and engaging with the academic debates that relate to the concept. What is your work adding to that? I think it is but needs to be reiterated throughout and given focused attention in the literature review.

  • We have now inserted a stand-alone section within the literature review on social connections and integration. We have re-centred our argument to demonstrate that care, resistance and connection are all contributions to integration, understood as a relational process that has affective as well as functional aspects.

The section on UK policy context required some more info. It would be useful for readers to learn about the Hostile Environment, for example, and the broader climate vis-à-vis immigration in the UK. This isn’t a huge addition but would be useful for those not familiar with the UK context and changing debates re. immigration and policies on refugees and people seeking asylum.

  • We have now added a new paragraph on the policy section to explain the UK hostile environment policy and to problematise this. We have also further linked to our findings and discussion to the hostile environment.

As a reader, I found the rich qualitative quotes more informative and useful when compared to the bar graphs. I would consider removing these as I’m not sure they are adding quite as much to the paper. A more minor issue: Does AMIF need more careful introduction and explanation?

  • The non-anonymised version of this paper includes a footnote explaining AMIF and the project which was funded, but this was removed from the version sent to reviewers to avoid identifying the authors. We have now changed this into project partners in the text and figures. We have chosen to retain the figures, as these have informed our analysis decision to focus on refugee-sector organisations, as opposed to the other connections identified as important by our participants.

On page 8 you mention that your findings could be considered biased. I would avoid this language. You have used a qualitative methodology for good reason and as you say, you can’t make sweeping generalisations but certain patterns can be identified. This is the nature of qualitative research so I wouldn’t concede quite so much here. Perhaps you could talk about your approach to sampling a little more carefully instead.

  • We have reviewed and expanded our section on sampling to address this comment, and have removed ‘biased’ instead noting that we cannot generalise our findings to the refugee population in Scotland. We have also briefly problematised how our sampling may have affected the connections which were most frequently mentioned during the focus groups.

As stated above, the quotes within the findings sections were really interesting and rich. It was great to see an attempt to bring the accounts of service providers/staff members and participants together. However, at the moment these insights are rather isolated and not brought back into dialogue with existing debates about integration in the asylum/immigration literature.

  • The discussion and conclusion has now been revised to make a clearer contribution to the existing literature on immigration policy and integration.

The paper could be more critical about the reasons for this level of support/care being provided by third sector organisations. Does this reflect a failing of the state and state policies (or just, quite simply, its absence) in relation to integration programmes? So, a bit more critical commentary on the broader political, economic conditions that have got us to this situation would be really good to see.

  • In our literature review, we outline some of the difficulties facing third sector organisations, linking these to immigration policy as well as to austerity and more generally to cutbacks to the welfare state. We now reflect back on these in more depth in the discussion and conclusion. 

Finally, you state that the paper goes some way to addressing the gap in research, but to what end. This is your opportunity to push why this is important. You filled the gap and what has that enabled – what has that enabled us to see better/understand?

  • We have now more clearly articulated our contribution through an overarching narrative that focuses on the three ways in which refugee sector organisations contribute to integration: through connecting, caring and resisting. This in turn, we suggest, highlights elements of integration that conceptually and empirically remain under-explored.

Round 2

Reviewer 1 Report

This manuscript has been much improved from its first iteration.  Absent additional research which would shore up its acknowledged empirical limitations, it has an original contribution to make in its present form, and thus I recommend publication.

Author Response

We would like to thank the reviewer for their positive feedback. This submitted version has been further proofread, with minor changes made to address specific points made by Reviewer 2. These have been marked with highlight to distinguish these from the tracked changes made to the previous version.

Reviewer 2 Report

Note p3 - only in Scotland and Wales is there agreement that other newcomers should be supported to integrate. Is Glasgow still the only dispersal city in Scotland? Ndofor-Tah is the lead author of the IoI and works for the Home Office - if you read the introduction etc you will see that this IoI is produced by and for the HO and not solely endorsed by it. Language needs changing therein. Re the issue of the 10 excluded interviews many will ask why not write the paper when you have analysed all data? Had you reached saturation? Its just not clear why timing is an issue - write the paper at the end of the study!

Figures are so much better! One the whole the whole paper is improved. It now needs a very good edit.

Author Response

We would like to thank the reviewer for their positive feedback. This submitted version has been further proofread. Changes made to respond to the specific points of feedback have been marked with highlight to distinguish these from tracked changes made to the previous version. We’ve addressed the specific points made by the reviewer in the attached.
